# Factors associated with infants' sunlight exposure among mothers attending the EPI unit of Wolkite University Specialized Hospital

**Girma Teferi Mengistu**[1]*, **Ayana Benti Terefe**[1], **Tolesa Gemeda Gudeta**[1], **Bizunesh Kefale Mengistu**[2]

**1** Department of Nursing, College of Medicine and Health Science, Wolkite University, Wolkite, Ethiopia,
**2** Department of Statistics, College of Natural and Computational Science, Ambo University, Ambo, Ethiopia

* girmateferi56@gmail.com, girma.teferi@wku.edu.et

**Data Availability Statement:** All relevant data are within the paper and its Supporting Information files.

## Abstract

### Background

Infant sunlight exposure in their early infancy is essential for the cutaneous synthesis of vitamin D. Vitamin D deficiency is highly prevalent due to inadequate exposure to sunlight. In Ethiopia, one in thirteen children had rickets, which could be prevented by sunlight exposure or supplementation. This study aimed to identify the practice of infants' sunlight exposure and associated factors among mothers attending the Expanded program immunization unit of Wolkite University Specialized Hospital.

### Method

This study employed an institutional-based cross-sectional study design to collect data. Data were collected from 220 mothers using an interviewer-administered questionnaire. The study employed a systematic random sampling technique to reach the study participants. The collected data were entered into a computer using the software Epidata 3.1 version and exported to SPSS version 23 for statistical analysis. Descriptive and inferential analysis was conducted. Logistic regression analysis was done, and a statistical association was declared at a p-value < 5% and a 95% confidence interval(CI). Then the results were presented using a frequency table, figures, and description.

### Result

A total of 220 infant-coupled mothers who visited the Expanded program immunization unit were included in the study. According to the current study, 67.3% of mothers had good practice with infants' sunlight exposure. Mothers' practice of infant sunlight exposure was affected by age of mothers 30–34 years, [AOR = 3.10, 95%CI (1.13, 8.51)], and age ≥35 years, [AOR = 4.49, 95%CI (1.20, 16.86)], and living in urban, [AOR = 1.94, 95%CI (1.053, 3.57)].

**Funding:** The author(s) received no specific funding for this work.

**Competing interests:** The authors have declared that no competing interests exist.

## Conclusion

The current study showed that two-thirds (67.3%) of mothers had good practice of sunlight exposure to their infants. Factors that affect mothers' practice of infants' sunlight exposure are age and place of residence. Health professionals should provide health education for mothers on the benefits of infants' sunlight exposure.

## Introduction

Infant sunlight exposure in their early infancy benefits in preventing diseases that could occur in their childhood development [1]. Sunlight exposure is essential for the cutaneous synthesis of vitamin D [2]. It is produced in the skin through a photosynthetic reaction from exposure to ultraviolet B (UVB) radiation. But clothing, excess body fat, sunscreen, and skin pigment melanin reduce the amount of the rays penetrating the skin [3]. The production of vitamin D less than body requirements can affect the health of infants [4].

Vitamin D is a fat-soluble vitamin recommended for exclusively breastfeeding infants. They can gain it through supplementation or sunlight exposure which produces it from the body [5]. The human body can generate about 80% of the vitamin D required for the body through sunlight exposure [6]. Vitamin D Supplementation for all breastfeeding infants also enables them to get the required amount [7].

Vitamin D prevents rickets in growing children and osteomalacia in adults [8]. The amount of vitamin D in the body determines the health of the bones [9]. Vitamin D can also prevent respiratory diseases such as wheezing [10].

The reasons for vitamin D deficiency are diet, atmospheric pollution, religious practices, geographical latitude and altitude, season, and time of the day [8]. Additionally, deficiency in mothers, older infants, and children with dark skin color, cultural practices, prolonged breast-feeding, and restriction of sunlight exposure are risk factors for its deficiency. In pediatrics, vitamin deficiency results in complications including hypocalcemia, seizure, rickets, limb pain, and fracture-related with poor bone formation [11,12].

Vitamin D deficiency is common among breastfeeding infants in sunny environments due to the avoidance of sunlight exposure and low supplementation [13]. Primary vitamin D deficiency is highly prevalent in countries with abundant sunshine due to inadequate sunlight exposure [14]. Mothers need awareness of the long-term effect of sunlight exposure on the growth and development of their infants [15].

Sunlight exposure initiates the body to produce vitamins due to sunlight rays. Exposure of infants starting from 6 weeks for 30 minutes per week, which lasts for 16 to 18 weeks, can achieve sufficient vitamin D at six months of age [16]. It prevents vitamin D deficiency-related bone disease, increased risk of respiratory illness, and low birth weight [17].

Adequate supplementation of vitamin D and calcium in infants can prevent nutritional rickets [18]. Calcium is supplied by the calcitriol hormone and used as a building material for bones. Sunlight exposure produces vitamin D in the skin, which in turn helps to produce calcitriol hormone [12]. In African countries, calcium intake is below the recommendations [19]. Rickets is the overt manifestation of vitamin D deficiency. If the infants are exclusively breastfed, they are prone to develop rickets [20]. Vitamin D deficiency is a public health issue in some low and middle-income countries. The risky groups in these countries are pregnant mothers, women of childbearing age, infants, and children [21].

Inadequate sunlight exposure is the cause of rickets in children [22]. In Ethiopia, the major causes of nutritional rickets in infants include lack of exposure to sunshine or inadequate

intake of vitamin D. The lack of infants' sunlight exposure is due to the traditional beliefs that prevent infants from exposure to sunlight [23].

Studies reported that children unexposed to sunlight had a deficiency of vitamin D and rickets [24]. According to a study conducted in central Ethiopia, the prevalence of vitamin D deficiency was 42% in school children [25]. On the other hand, a study conducted in eastern Ethiopia identified that one in thirteen, 7.8% of children had rickets [26]. Similarly, a study conducted at Jimma University Specialized Hospital reported that the prevalence of rickets was 10.5% in children [22]. But, sunlight exposure can treat deficiency-related rickets within weeks [27].

Previously conducted studies identified that the practice of infants' sunlight exposure ranges from 44.6% to 58% [28–32]. Among these studies, only three of them studied associated factors where variables like maternal education, husband education, mother occupation, family size, and fear of cold were determining factors of infants' sunlight exposure [28,30,31]. The current study included the place of residence in addition to previously studied variables. In Ethiopia, there is a cultural ceremony that prevents mothers from being outdoors during the postnatal period [33].

Ethiopia has access to sunlight due to its geographical location; however, the community does not expose its infants to sunlight, and there is no nutritional supplementation of the vitamin. In Ethiopia, a cultural practice restricts mothers and infants indoors after delivery during the early weeks. In the absence of sunlight exposure with no supplementation of vitamin D, infants are risky to develop vitamin D deficiency-related complications. Therefore, this study aimed to assess the practice of sunlight exposure and associated factors among mothers attending the Expanded program on immunization (EPI) at Wolkite University Specialized Hospital.

## Materials and methods

### Study area, design, and period

The study was conducted at Wolkite University Specialized Hospital (WUSH) using an institutional-based cross-sectional study design from 01 June to 30 July 2021. WUSH is 158km from Addis Ababa to the south in the Gurage zone, Southern Nation, Nationalities, and Peoples Region. The hospital serves as a teaching and referral hospital. Currently, it is providing emergency services, outpatients' services, inpatients services, major and minor surgeries, maternal care, pediatrics services, antiretroviral therapy (ART) and Tuberculosis (TB) treatment clinics, laboratory, radiography services, and EPI services for the community.

### Source and study population

The source population was all mothers attending the EPI unit of WUSH for immunization service. The study population was all mothers with infants who come to WUSH to vaccinate their infants and who fulfill the inclusion criteria.

### Inclusion and exclusion criteria

The study included mothers with up to a one-year baby who visited the EPI unit of WUSH. The study excluded mothers with critically ill babies and mental or physical disabilities.

### Sample size determination

The sample was calculated using a 95% confidence interval level, a 5% margin of error, and a 54.5% proportion of neonatal sunlight exposure from a study conducted at Yirgalem General Hospital [30]. Based on the given information the sample was 381. On average, 420 mothers

attended the EPI unit of WUSH in two months to get immunization services for their infants. We used the correction formula to get the sample needed for the study. Then the corrected sample size was 200 and adding a 10% non-response rate the final sample size was 220.

## Sampling procedure

A systematic random sampling technique was employed to select study participants. On average, an estimated 420 mothers visited the EPI unit of the hospital for immunization services in two months. Then the sampling interval (k) was calculated by dividing the number of infants served within two months by the sample size. This was calculated as k = N/n, 420/220 = 1.9. Approximately the k was 2. The data was collected from every other mother. During the data collection time, the first mother was selected randomly by the lottery method.

## Operational definition

**Good knowledge.** Respondents who scored more than the mean value of responses for knowledge-related questions. Good practice: Respondents who scored more than the mean value of responses for practice-related questions.

## Data collection

Data were collected using an interviewer-administered semi-structured questionnaire. The questionnaire was adapted from other previous studies [28,29,30,32]. The questionnaire had divided into four components which include sociodemographic characteristics (nine questions), obstetric history of mothers (five questions), knowledge questions (seven questions), and practice questions (eleven questions). The questionnaire was pretested on 5% (11) of mothers attending the EPI unit of Atat Hospital. The pretest was done to check the clarity, completeness, skip pattern, and order of questions, and then we made a minor modification. Experts revised the questionnaire before data collection and the data were collected by three BSc nurses after one-day training on the objective, purpose, and confidentiality of the study. The principal investigator supervised data collection data collection precess to increase data quality.

## Data analysis procedure

After data collection, the data were checked for completeness and then entered into a computer using software; Epidata 3.1 version. Then we exported the data to software; SPSS version 23. After exporting the data, we conducted descriptive and inferential analysis. The descriptive part was used to analyze the data using frequency tables with descriptions. The inferential analysis was used to analyze the relation of independent variables with dependent variables, infants' sunlight exposure. Variables with a p-value of less than 0.2 in bivariate logistic regression analyses were selected for multivariable logistic regression analysis. We conducted multivariable logistic regression analysis at a p-value < 5%, and 95%CI and a variable with a p-value of less than 0.05 were declared statistically significant. The inferential analysis included bivariate and multivariable logistic regression analyses. The result of the logistic regression analysis was presented using the Adjusted odds ratio (AOR) at 95%CI.

## Ethical consideration

The research committee of the Nursing Department of Wolkite University revised and approved the study. The department of Nursing gave us a letter of permission during data

collection. We obtained informed consent from each respondent after explaining the purpose and objective of the study.

## Results

### Sociodemographic characteristics

Data were collected from 220 mothers attending the immunization unit, which gives a 100% response rate. Among the mothers who participated in the study, 110(50%) were in the age group of 25–29. More than three fourth, 175 (79.5%) of the infants were six weeks or less during data collection. One-third (34.3%) of mothers attended primary education. About two-thirds (67.7%) of mothers were from urban areas (Table 1).

**Table 1. Sociodemographic characteristics of a mother attending the EPI unit at WUSH, 2021.**

| Variable | Categories | Response | |
|---|---|---|---|
| | | Frequency | Percent |
| Age of mother | 20–24 | 22 | 10.0 |
| | 25–29 | 110 | 50.0 |
| | 30–34 | 66 | 30.0 |
| | ≥35 | 22 | 10.0 |
| Age of infant(in months) | 0–6 | 175 | 79.5 |
| | 7–12 | 45 | 20.5 |
| Marital status | Single | 2 | 0.9 |
| | Married | 216 | 98.2 |
| | Widowed | 2 | 0.9 |
| Mother's educational status | no formal education | 38 | 17.3 |
| | primary education | 75 | 34.1 |
| | secondary education | 62 | 28.2 |
| | college/diploma | 45 | 20.5 |
| Occupation of mother | Student | 3 | 1.4 |
| | Housewife | 128 | 58.2 |
| | Daily laborer | 14 | 6.4 |
| | Government employee | 33 | 15.0 |
| | Private employee | 11 | 5.0 |
| | Merchant | 31 | 14.1 |
| Place of residence | Urban | 149 | 67.7 |
| | Rural | 71 | 32.3 |
| Family size | 1–3 | 60 | 27.3 |
| | 4–6 | 156 | 70.9 |
| | ≥7 | 4 | 1.8 |
| Monthly income (in ETB) | ≤1800 birr | 16 | 7.3 |
| | 1801–3800 birr | 141 | 64.1 |
| | 3801–7500 birr | 60 | 27.3 |
| | >7501 birr | 3 | 1.4 |
| Husband educational status | No formal education | 25 | 11.4 |
| | Primary education | 63 | 28.6 |
| | Secondary education | 82 | 37.3 |
| | College and above | 50 | 22.7 |

WUSH = Wolkite University Specialized Hospital, EPI = Expanded program of immunization.

**Table 2. Obstetric characteristics of mothers attending EPI unit of WKUSH, 2021.**

| Variable | Characteristics | Frequency | Percentage |
|---|---|---|---|
| Number of ANC visits during pregnancy | <4 | 90 | 40.9 |
| | 4 | 130 | 59.1 |
| Place of delivery | Home | 1 | 0.5 |
| | Health center | 114 | 51.8 |
| | Hospital | 105 | 47.7 |
| Gestational age | <37 | 74 | 33.6 |
| | 37–42 | 145 | 65.9 |
| | >42 | 1 | 0.5 |
| Birth weight | <2.5 | 17 | 7.7 |
| | ≥2.5 | 203 | 92.3 |

WUSH = Wolkite University Specialized Hospital, EPI = Expanded program of immunization.

## Obstetric characteristics of mothers

All respondents had ANC follow-ups during their last recent pregnancy, of which 130 (59.1%) had four visits. About half (51.8%) of mothers gave birth at a health center. One-third (33.6%) of mothers gave birth at less than 37 weeks. Regarding birth weight, 17(7.7%) of neonates were less than 2500 grams during delivery (Table 2).

## Knowledge of mothers about infants' sunlight exposure

Almost all mothers (98.2%) had information about infants' sunlight exposure. However, about half of them (51.4%) said that infants' sunlight exposure had harmful effects to infants if improperly exposed. More than two-thirds (99.1%) of the mothers said that morning is a good time for infants' sunlight exposure (Table 3). The overall knowledge of sunlight exposure of infants among mothers was 51.8% (Fig 1).

## Practices of infants' sunlight exposure

Out of 220 study participants, 162(73.6%) mothers exposed their infants to sunlight. Of these, 146 (90.1%) started sunlight exposure after two weeks of delivery. About 111 (68.5%) of the mothers exposed their infants to sunlight daily. Almost two-thirds (63.6%) of mothers exposed their babies to sunlight outdoors. More than ninety percent (93.2%) of the mothers exposed their infants to sunlight in the morning, 8–10 AM. Fifty percent (50.6%) of them covered their infants partially during sunlight exposure. Eighty-three percent (83.3%) of mothers exposed their infants for 15–30 minutes. More than ninety percent (92.0%) of the respondents applied lubricants during the exposure (Table 4). Two-thirds (67.3%) of mothers had good practices of infants' sunlight exposure which is computed based on the mean value of their responses to the practice questions (Fig 2).

## Factors associated with sunlight exposure of infants

Logistic regression was done to identify a statistically significant variable at a p-value of < 5% and a 95%CI. Both bivariate and multivariable logistic regression analyses were done. In bivariate logistic regression analysis, the age of mothers, place of residence, and family size were significantly associated with mothers' practice of infants' sunlight exposure. We used the Crude odds ratio (COR) for bivariate logistic regression analysis. Mothers in the age group of 30–34 years and ≥35 were 3.20 [COR = 3.20, 95%CI (1.18, 8.69)], and 4.08 [COR = 4.08, 95%CI (1.11, 15.02)] times more likely to expose their infants to sunlight than mothers in the age

**Table 3. Knowledge of mothers about infants' sunlight exposure among mothers attending EPI unit at WUSH, 2021.**

| Variable | Characteristics | Frequency | Percent |
|---|---|---|---|
| Heard about sunlight exposure | Yes | 216 | 98.2 |
| | No | 4 | 1.8 |
| Source of information | Health professionals | 155 | 70.5 |
| | Media | 9 | 4.1 |
| | Neighbors | 54 | 24.5 |
| | Friends | 75 | 34.1 |
| Sunlight exposure is beneficial | Yes | 218 | 99.1 |
| | No | 2 | 0.9 |
| Benefits | Strengthen bone | 147 | 67.4 |
| | Strengthen teeth | 7 | 3.2 |
| | Keep child warm | 30 | 13.8 |
| | Produce vitamin D | 70 | 32.1 |
| | Strengthen body | 140 | 64.2 |
| Good time to expose | Morning | 218 | 99.1 |
| | Afternoon | 2 | 0.9 |
| Does sunlight exposure is harmful if not exposed appropriately | Yes | 114 | 51.4 |
| | No | 106 | 48.2 |
| Harms they fear | Skin cancer | 92 | 80.7 |
| | Sterility | 1 | 0.9 |
| | Blindness | 33 | 28.9 |

WUSH = Wolkite University Specialized Hospital, EPI = Expanded program of immunization.

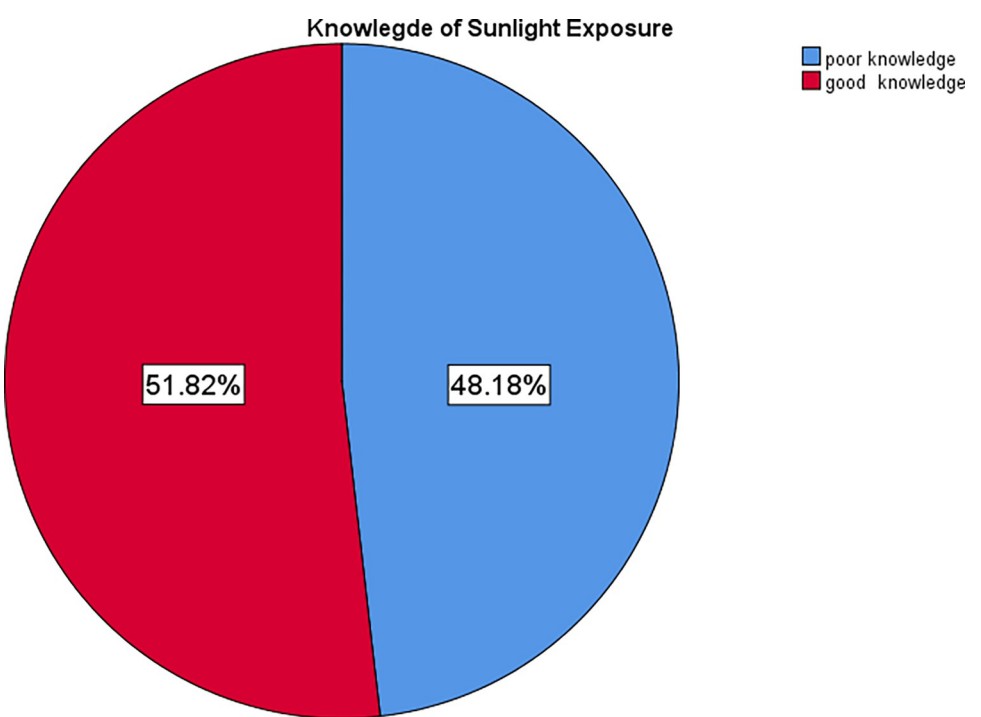

**Fig 1. Knowledge of infants' sunlight exposure among mothers attending EPI unit at WUSH, 2021.**

**Table 4. Practice infants' sunlight exposure among mothers attending the EPI unit at WUSH, 2021.**

| Variable | Category | Frequency | Percent |
|---|---|---|---|
| Do you expose your infant to sunlight | Yes | 162 | 73.6 |
| | No | 58 | 26.4 |
| Age of infant to start sunlight exposure | ≤ 7days | 1 | 0.6 |
| | 8 -14days | 15 | 9.3 |
| | ≥15days | 146 | 90.1 |
| How frequently do you expose | Daily | 111 | 68.5 |
| | Sometimes | 51 | 31.5 |
| Do you expose your infant to sunlight outdoors? | Yes | 103 | 63.6 |
| | No | 59 | 36.4 |
| At what time do you expose | Morning 8–10 am | 151 | 93.2 |
| | Mid-day 11 am -1pm | 7 | 4.3 |
| | Afternoon 2-4pm | 4 | 2.5 |
| Condition of clothing during exposure | Unclothed | 28 | 17.3 |
| | With diaper and eye protection | 39 | 24.1 |
| | Partially covered | 82 | 50.6 |
| | Completely covered | 13 | 8.0 |
| How many minutes do you expose your neonate | <10 minute | 26 | 16.0 |
| | 15–30 minute | 135 | 83.3 |
| | >30 | 1 | 0.6 |
| Apply lubricant during exposure | Yes | 149 | 92.0 |
| | No | 13 | 8.0 |
| When do you apply | Before exposure | 25 | 16.8 |
| | During exposure | 25 | 16.8 |
| | After exposure | 99 | 66.4 |
| What thing do you apply | Baby Vaseline | 86 | 57.7 |
| | Baby lotion | 31 | 20.8 |
| | Butter | 32 | 21.5 |
| Fear to expose baby to sunlight (n = 149) (Multiple responses are possible) | Sickness | 66 | 44.3 |
| | Evil eye | 87 | 58.4 |
| | Cold | 65 | 43.6 |

WUSH = Wolkite University Specialized Hospital, EPI = Expanded program of immunization.

group of 20–24 respectively. Mothers from the urban area were 1.86 [COR = 1.86, 95%CI (1.03, 3.37)] times more likely to expose their infants to sunlight than mothers from rural areas. Mothers from 4–6 families were 2.08 [COR = 2.08, 95%CI (1.12, 3.86)] times more likely to expose their infants to sunlight than mothers from 1–3 families.

Multivariable logistic regression analysis was done, and two variables (age of mothers and place of residence) were statistically significant. Mothers in the age group of 30–34 years and ≥35 were 3.10 [AOR = 3.10, 95%CI (1.13, 8.51)] and 4.49 [AOR = 4.49, 95%CI (1.20, 16.86)] times more likely to expose their infants to sunlight than mothers in the age group of 20–24 respectively. Mothers from urban areas were 1.94 [AOR = 1.94, 95%CI (1.05, 3.57)] times more likely to expose their infants to sunlight than mothers from rural areas (**Table 5**).

## Discussion

The current study identified that 98.2% of the respondents heard about infants' sunlight exposure. Among those who heard about infants' sunlight exposure, 70.5% of the study participants

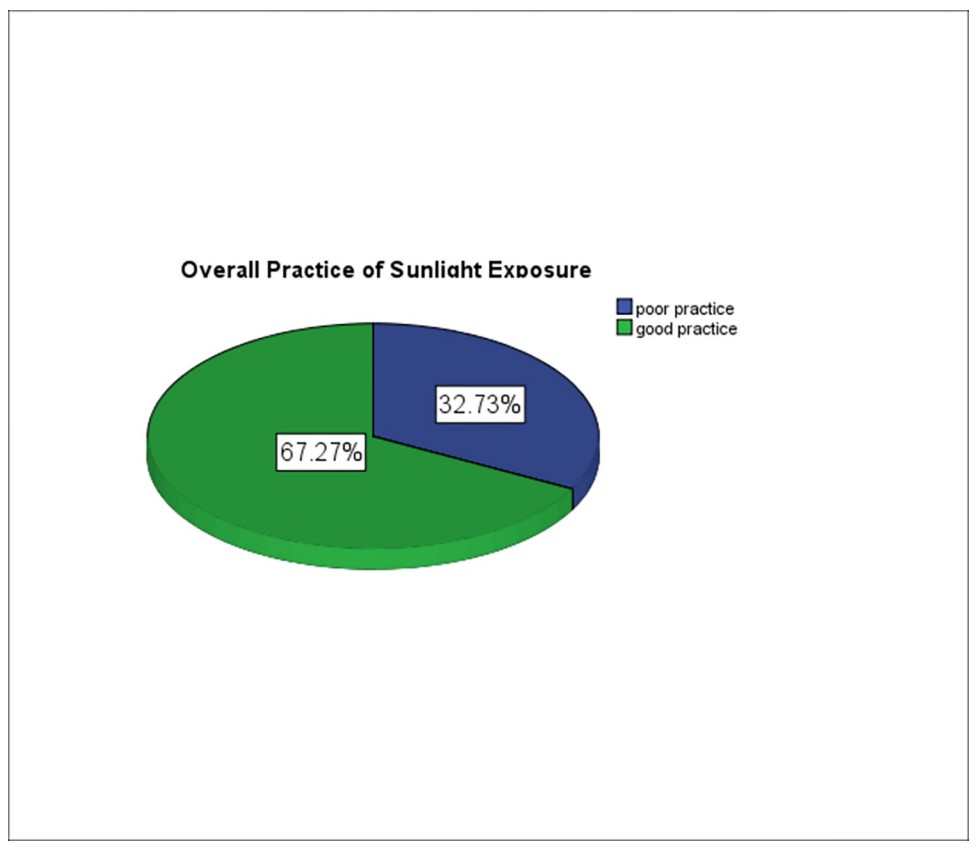

**Fig 2. Practice of infants' sunlight exposure among mothers attending EPI unit at WUSH, 2021.**

heard from health professionals. In the case of infants' sunlight period, 99.1% of them knew that infants' sunlight exposure is in the morning. Among the study participants, only 51.8% of them had good knowledge of infants' sunlight exposure.

The finding of the study revealed that 73.6% of respondents have exposed their infants to sunlight. among mothers who exposed their infants to sunlight, 90.1% of them started

**Table 5. Factors associated with sunlight exposure of infants among mothers attending EPI unit at WUSH, 2021.**

| Variable | Category | Practice of sunlight exposure | | COR:95% CI | AOR:95% CI |
|---|---|---|---|---|---|
| | | Good practice | Poor practice | | |
| Age of mothers | 20–24 | 10(45.5%) | 12(54.5%) | 1 | 1 |
| | 25–29 | 73(66.4%) | 37(33.6%) | 2.37(0.94, 5.99) | 2.20(0.86, 5.65) |
| | 30–34 | 48(72.7%) | 18(27.3%) | 3.20(1.18, 8.69)* | 3.10(1.13, 8.51)* |
| | 35+ | 17(77.3%) | 5(22.7%) | 4.08(1.11,15.02* | 4.49(1.12, 16.86)* |
| Residence | Urban | 107(71.8%) | 42(28.2%) | 1.86(1.03, 3.37)* | 1.94(1.05, 3.57)* |
| | Rural | 41(57.7%) | 30(42.3%) | 1 | 1 |
| Family size | 1–3 | 33(55.0%) | 27(45.0%) | 1 | 1 |
| | 4–6 | 112(71.8%) | 44(28.2%) | 2.08(1.12, 3.86)* | - |
| | ≥7 | 3(75.0%) | 1(25.0%) | 2.46(0.24, 24.97) | - |

* = p< 0.05.

WUSH = Wolkite University Specialized Hospital, EPI = Expanded program of immunization.

exposure after the age of two weeks of delivery. Among those mothers who exposed their infants, two-thirds of them exposed their infants daily to the sunlight. During infants' sunlight exposure, 50.6% of mothers exposed their infants to sunlight by partially covering their babies' bodies with a cloth. During infants' sunlight exposure, the mothers reported that they fear sickness, evil eyes, and cold.

Our study reveals that 67.3% of mothers had good practice in infants' sunlight exposure. This finding was higher than the study finding from Debre Markos 44.6% [28], Ferta District 45.7% [29], Yirgalem Hospital 54.5% [30], Aleta Wendo Health Center 58% [32], and St. Paul's Hospital 54. 6% [31]. The higher finding of the current study could be related to the time gap between the studies and study area. The mothers received advice on infants' sunlight exposure, and the current study was conducted at a specialized hospital.

Logistic regression analysis was done to identify the factors associated with infants' sunlight exposure among the study participants. In the bivariate analysis, there were three variables statistically significantly associated with mothers' practice of infants' sunlight exposure. The variables were the age of mothers, place of residence of mothers, and family size. Similarly, a study conducted in Debre Markos town showed that the age of mothers and family size were significantly associated [28].

The multivariable logistic regression analysis showed that the age of mothers and place of residence were statistically significant. Mothers in the age group of thirty to thirty-four and thirty-five and above were 3 and 4.5 times more likely to expose their infants to sunlight than mothers in the age group of twenty to twenty-four respectively. Mothers from urban areas are 1.94 times more likely to expose infants to sunlight than mothers from rural areas. Family size was significantly associated in a study conducted at Debre Markos town [28], while it was significantly associated in bivariate logistic regression analysis in the current study. Mothers at young age and mothers living in rural areas had a poor practice of infants' sunlight exposure. These groups of mothers need attention to create awareness among them.

We addressed the place of residence of study participants, which was not studied in the previous studies. The study was based on mothers' responses regarding their practice of infants' sunlight exposure, and no sample was studied from infants to assess the concentration of vitamin D. Despite these limitations, this study added important information about mothers' practice of infants' sunlight exposure.

## Conclusion

The findings of this study indicated that about two-thirds of mothers have good practice of infant sunlight exposure. Factors that affect mothers' practice of infants' sunlight exposure are age and place of residence. Health professionals should provide health education for mothers at young ages and mothers from rural areas on the benefits of infants' sunlight exposure.

## Supporting information

**S1 File.**
(SAV)

## Acknowledgments

We would like to thank Wolkite University Specialized Hospital and our study participants for their participation.

## Author Contributions

**Conceptualization:** Girma Teferi Mengistu, Ayana Benti Terefe, Tolesa Gemeda Gudeta.

**Data curation:** Girma Teferi Mengistu, Tolesa Gemeda Gudeta, Bizunesh Kefale Mengistu.

**Formal analysis:** Bizunesh Kefale Mengistu.

**Investigation:** Girma Teferi Mengistu, Tolesa Gemeda Gudeta.

**Methodology:** Girma Teferi Mengistu, Ayana Benti Terefe, Tolesa Gemeda Gudeta, Bizunesh Kefale Mengistu.

**Project administration:** Girma Teferi Mengistu.

**Resources:** Girma Teferi Mengistu, Ayana Benti Terefe.

**Supervision:** Girma Teferi Mengistu, Ayana Benti Terefe.

**Validation:** Bizunesh Kefale Mengistu.

**Visualization:** Ayana Benti Terefe.

**Writing – original draft:** Girma Teferi Mengistu.

**Writing – review & editing:** Tolesa Gemeda Gudeta, Bizunesh Kefale Mengistu.

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

     34801045

