## [Decision Letter · Decision Letter 0]

15 Jun 2022

PONE-D-21-36839Factors Associated With Infants’ Sunlight Exposure among Mothers Attending EPI of Wolkite University Specialized HospitalPLOS ONE

Dear Dr. Mengistu,

Thank you for submitting your manuscript to PLOS ONE. After careful consideration, we feel that it has merit but does not fully meet PLOS ONE’s publication criteria as it currently stands. Therefore, we invite you to submit a revised version of the manuscript that addresses the points raised during the review process.

I would like to sincerely apologise for the delay you have incurred with your submission. It has been exceptionally difficult to secure reviewers to evaluate your study. We have now received two completed reviews; the comments are available below. The reviewers have raised several significant scientific concerns about the study that need to be addressed in a revision.

Please revise the manuscript to address all the reviewer's comments in a point-by-point response in order to ensure it is meeting the journal's publication criteria. Please note that the revised manuscript will need to undergo further review, we thus cannot at this point anticipate the outcome of the evaluation process.

We look forward to receiving your revised manuscript.

Kind regards,

Miquel Vall-llosera Camps

Senior Editor

PLOS ONE

Journal Requirements:

please request copy editing. Do not ping with follow up.

2. Please include a separate caption for each figure in your manuscript.

Reviewers' comments:

Reviewer's Responses to Questions

**Comments to the Author**

1. Is the manuscript technically sound, and do the data support the conclusions?

Reviewer #1: Partly

Reviewer #2: Partly

2. Has the statistical analysis been performed appropriately and rigorously? 

Reviewer #1: Yes

Reviewer #2: No

3. Have the authors made all data underlying the findings in their manuscript fully available?

Reviewer #1: Yes

Reviewer #2: Yes

4. Is the manuscript presented in an intelligible fashion and written in standard English?

Reviewer #1: No

Reviewer #2: Yes

5. Review Comments to the Author

Reviewer #1: Dear Authors....

This study focusing on the “ Factors Associated With Infants’ Sunlight Exposure among Mothers Attending EPI of Wolkite University Specialized Hospitals” is not uncommon, but the culture and understanding level about Infants’ Sunlight Exposure varies from location to location and culture to culture within a country and the authors used the population in Gurage , Southern-Ethiopia, which is stimulating to the readers. Authors requested to perform a major revision including language correction. I try to highlight some major and minor comments that the authors could reflect and thus confidently improving the manuscript even further.

Abstract part page-2

Q1. line-26 EPI please! change to Expanded program on immunization(EPI)

Method part page-2

*please! write the abstract part properly and clearly because it is a mirror of your manuscript including sampled number of population, study design and sampling technique

Q2. what was the study design?

Q3. what was the sampling techniques?

Result part page-2

Q4. line-32. how many infant coupled mothers were recruited in your study? please! start like A total of ........infant coupled mothers who visited EPI for vaccinating their children were included in the study.

Conclusion part page-2

Q5. Your conclusion is more than your result and please! revise it properly

Introduction part page-3

Q6. line 49 UVB please! change to ultra violet B (UVB)

Q7. pge-5 line 107 EPI please! change to Expanded program on immunization(EPI)

Materials and methods part page 6

Q8. inclusion and exclusion part line 128: what was the age infants which were included in the study? please! mention properly in the inclusion part.

Q9. There was no any information on how authors selected 220 infant coupled mothers for this study? no sample size estimation information? is it single/two population proportion formula? purposive?

Q10. ......knowledge, and practice questions. Every day, the filled questionnaire was

checked for completeness by the principal investigator to increase data quality. How authors categorized the knowledge level practice level of of mothers? or nothing was defined about level of knowledge and practice? nothing was mentioned in the manuscript. How many Likert scaled questions were applied? Have you use principal component analysis? or threshold formula or mean/median score?

Q11. result part page 9; Please! remove ANC visit rows from table two because it doesn't give any meaning

Mothers’ knowledge about sunlight exposure of neonates part page-9

Q12. line 183-184. The overall knowledge of sunlight exposure of infants among mothers was 51.8%. How did you get this proportions? what was your analytic approach to calculate this proportion?

The practice of Sunlight Exposure part page 11.

Q13. line 197-198 The overall practice of sunlight exposure among them were 67.3% during the study period. Similar to Q12.

Discussion part page-15.

In general the conclusion part was written poorly and the authors' requested to re-write this part by reviewing different literatures which were conducted in different parts with similar topic. please! the following literatures

1. https://www.dovepress.com/getfile.php?fileID=75617

2. https://bmcresnotes.biomedcentral.com/track/pdf/10.1186/s13104-019-4221-4.pdf

3. https://journals.sagepub.com/doi/pdf/10.1177/11795565211041348

4. https://www.hindawi.com/journals/ijrmed/2019/2638190/

Reviewer #2: The authors described the sunlight exposure among infants and the factors associated with it.

The authors could improve the manuscripts in the following sections:

Introduction:

1. The Introduction section needs to be summed up concisely.

2. The author may want to provide the prevalence of vitamin D deficiency among infants in the study area to show that sunlight exposure is very important, since “vD deficiency in order of magnitude was found in 62%, 28%, and 6% of Mexican, Cincinnati and Shanghai infants, respectively (p<0.001)”.

3. The logic flow of the introduction needs to be improved. For example, you need to provide the association between vitamin D and calcium before talking about “In African countries, calcium intake is below recommendations”.

4. The authors need to describe the importance/innovation of this study. Is there a knowledge gap in the research question? Is this study better than prior studies?

Method:

1. A comprehensive literature review is expected in the method section to identify the potential factors associated with sunlight exposure among infants from prior publications.

2. The authors may want to emphasize the representative of the study population when describing the study hospital.

3. The audience would like to read about sample size and power calculation in the method section.

4. Please provide more information about the design of the questionnaire and the validation of the questionnaire.

5. Please describe the training of interviewers before the study to improve the quality of the data collection.

Results:

1. The authors may want to add the “outdoor” to the sunlight in the questionnaire since indoor sunlight exposure may not lead to vitamin D synthesis.

2. The distribution of “sunlight is beneficial” and “sunlight is harmful” were conflict with each other in Table 3.

3. Please add the unit of monthly income in table 1.

4. The percentages mentioned in the Tables 3-4 were not consistent with the percentages mentioned in the result section.

5.“162(73.6%) of mothers exposed their infants to sunlight. Of these,146(66.4%) started sunlight exposure from 3weeks” (row 189) was not provided in table 4.

6. Please keep two decimal places for ORs.

7. Please describe the definition of the dependent variable for the regression model. The sample size for good practice is 148 in table 5, which is not consistent with any prior tables. The 148 is consistent with figure 2, but the audience still could not find the definition of “good practice” in the manuscript.

Discussion:

1. In the first paragraph of the discussion, please just emphasize the most important findings.

2. Please add strengths, limitations, biological mechanisms, and public health implications in the discussion section. For example, please discuss the potential selection bias and limited generalizability due to the use of “specialized” hospitals and the description of the sampling method.

Conclusion:

It is too early to provide public health policy recommendations based on one cross-sectional study among participants from one specialized hospital “Government should prepare a guideline on time of exposure and duration including starting age.”, “Each health institution has to work in providing health education for mothers during ANC visits, during discharge after delivery. Health education that could be given home to home by health extension workers for rural mothers. Government should prepare a guideline on time of exposure and duration including starting age.”

Grammar:

1. The following sentences need to be improved for readability:

1) “The practice was determined by the age of mothers and their residence. ”

2) “Compared with other previous studies used in discussion, there is progress from time to time even though the

study area was not the same.”

3) “Being in the age category of 30–34, 3.095 [AOR= 3.095, 95%CI (1.126, 8.506)], being in the age category of

35+ were 4.491 [AOR= 4.491, 95%CI (1.196, 16.863)], Living in urban 1.938 [AOR= 1.938, 95%CI (1.053,

3.566)] more likely to expose infants to sunlight than living in rural”

4) “Health education that could be given home to home by health extension workers for rural mothers.”

2. There are inconsistent capitalization, inappropriate use of abbreviations, and inconsistent fonts.

3. No reference for the sentence for rows 48,53, 57, and 77

4. There is repetitive “of clothing” in Table 4

6. PLOS authors have the option to publish the peer review history of their article (what does this mean?). If published, this will include your full peer review and any attached files.

Reviewer #1: **Yes: **Agete Tadewos Hirigo (MSc, Assistant professor of Clinical chemistry and Ph.D. fellow)

Reviewer #2: No

---

## [Author Response · Author response to Decision Letter 0]

1 Aug 2022

Responses to Editor(s) and Reviewers 

Manuscript Id: PONE-D-21-36839

Title: Factors Associated with Infants’ Sunlight Exposure among Mothers Attending EPI of Wolkite University Specialized Hospital

Dear editor(s) and Reviewers I would like to say thank for your comments and recommendations. I have included the concerns raised by reviewers in the documents, and if there is untouched points I am ready to amend. Thank you again. 

 Reviewer #1 

Section Comments Responses

Abstract part page-2

 Q1. line-26 EPI please! change to Expanded program on immunization(EPI) Changed 

 Method part page-2

*please! write the abstract part properly and clearly because it is a mirror of your manuscript including sampled number of population, study design and sampling technique Revised and the raised concerns are incorporated. 

 Q2. what was the study design? Included 

 Q3. what was the sampling techniques? Included 

 Result part page-2

Q4. line-32. how many infant coupled mothers were recruited in your study? please! start like A total of ........infant coupled mothers who visited EPI for vaccinating their children were included in the study. The result part is revised and modified. 

 Conclusion part page-2

Q5. Your conclusion is more than your result and please! revise it properly Revised and updated 

Introduction part page-3

 Q6. line 49 UVB please! change to ultra violet B (UVB) Changed 

 Q7. pge-5 line 107 EPI please! change to Expanded program on immunization(EPI) Changed 

Materials and methods part page 6

 Q8. inclusion and exclusion part line 128: what was the age infants which were included in the study? please! mention properly in the inclusion part. Included 

 Q9. There was no any information on how authors selected 220 infant coupled mothers for this study? no sample size estimation information? is it single/two population proportion formula? purposive? Sample size determination is included in the document. 

 Q10. ......knowledge, and practice questions. Every day, the filled questionnaire was checked for completeness by the principal investigator to increase data quality. How authors categorized the knowledge level practice level of of mothers? or nothing was defined about level of knowledge and practice? nothing was mentioned in the manuscript. How many Likert scaled questions were applied? Have you use principal component analysis? or threshold formula or mean/median score? An operational definition is added in the document for both knowledge and practice questions. 

Result Q11. result part page 9; Please! remove ANC visit rows from table two because it doesn't give any meaning Removed 

 Mothers’ knowledge about sunlight exposure of neonates part page-9 

Q12. line 183-184. The overall knowledge of sunlight exposure of infants among mothers was 51.8%. How did you get this proportions? what was your analytic approach to calculate this proportion? This is explained under the operational definition.

 The practice of Sunlight Exposure part page 11.

Q13. line 197-198 The overall practice of sunlight exposure among them were 67.3% during the study period. Similar to Q12. This is explained under operational definition.

Discussion part page-15.

 In general the conclusion part was written poorly and the authors' requested to re-write this part by reviewing different literatures which were conducted in different parts with similar topic. The conclusion is revised and updated.

 Reviewer #2 

Introduction: 1. The Introduction section needs to be summed up concisely. Rearranged 

 2. The author may want to provide the prevalence of vitamin D deficiency among infants in the study area to show that sunlight exposure is very important, since “vD deficiency in order of magnitude was found in 62%, 28%, and 6% of Mexican, Cincinnati and Shanghai infants, respectively (p<0.001)”. It is included in the document 

 3. The logic flow of the introduction needs to be improved. For example, you need to provide the association between vitamin D and calcium before talking about “In African countries, calcium intake is below recommendations”. It is added and rearranged 

 4. The authors need to describe the importance/innovation of this study. Is there a knowledge gap in the research question? Is this study better than prior studies? Included in the document, in the colored part of the literature review 

Method: 1. A comprehensive literature review is expected in the method section to identify the potential factors associated with sunlight exposure among infants from prior publications. In reviewing the submission guideline, I found that a comprehensive literature review is under the introduction part. I added it at the end of the introduction part which is highlighted. 

 2. The authors may want to emphasize the representative of the study population when describing the study hospital. Even though the Hospital is a specialized hospital, it is surrounded by rural villages where it serves them in addition to referral cases. 

 3. The audience would like to read about sample size and power calculation in the method section. Included 

 4. Please provide more information about the design of the questionnaire and the validation of the questionnaire. Included 

 5. Please describe the training of interviewers before the study to improve the quality of the data collection. Included 

Results: 1. The authors may want to add the “outdoor” to the sunlight in the questionnaire since indoor sunlight exposure may not lead to vitamin D synthesis. Added 

 2. The distribution of “sunlight is beneficial” and “sunlight is harmful” were conflict with each other in Table 3. Revised 

 3. Please add the unit of monthly income in table 1. Added 

 4. The percentages mentioned in the Tables 3-4 were not consistent with the percentages mentioned in the result section. Sorry, it was a mistake done during descriptive analysis. Rather than using Valid percent, we used the normal percent. Now the table and the descriptions are revised and updated. 

 5.“162(73.6%) of mothers exposed their infants to sunlight. Of these,146(66.4%) started sunlight exposure from 3weeks” (row 189) was not provided in table 4. 

 6. Please keep two decimal places for ORs. Changed 

 7. Please describe the definition of the dependent variable for the regression model. The sample size for good practice is 148 in table 5, which is not consistent with any prior tables. The 148 is consistent with figure 2, but the audience still could not find the definition of “good practice” in the manuscript. Included 

Discussion:

 1. In the first paragraph of the discussion, please just emphasize the most important findings. Addressed 

 2. Please add strengths, limitations, biological mechanisms, and public health implications in the discussion section. For example, please discuss the potential selection bias and limited generalizability due to the use of “specialized” hospitals and the description of the sampling method. Revised and updated 

Conclusion It is too early to provide public health policy recommendations based on one cross-sectional study among participants from one specialized hospital “Government should prepare a guideline on time of exposure and duration including starting age.”, “Each health institution has to work in providing health education for mothers during ANC visits, during discharge after delivery. Health education that could be given home to home by health extension workers for rural mothers. Government should prepare a guideline on time of exposure and duration including starting age.” Revised and updated 

Grammar:

 1. The following sentences need to be improved for readability:

1) “The practice was determined by the age of mothers and their residence.” 

2) “Compared with other previous studies used in discussion, there is progress from time to time even though the study area was not the same.” 

3) 3) “Being in the age category of 30–34, 3.095 [AOR= 3.095, 95%CI (1.126, 8.506)], being in the age category of 35+ were 4.491 [AOR= 4.491, 95%CI (1.196, 16.863)], Living in urban 1.938 [AOR= 1.938, 95%CI (1.053, 3.566)] more likely to expose infants to sunlight than living in rural” 

4) “Health education that could be given home to home by health extension workers for rural mothers.” Revised and updated 

 2. There are inconsistent capitalization, inappropriate use of abbreviations, and inconsistent fonts. Revised 

 3. No reference for the sentence for rows 48,53, 57, and 77 Revised 

 4. There is repetitive “of clothing” in Table 4 Revised

---

## [Decision Letter · Decision Letter 1]

2 Oct 2022

PONE-D-21-36839R1Factors associated with infants’ sunlight exposure among mothers attending the EPI unit of Wolkite University Specialized HospitalPLOS ONE

Dear Dr. Mengistu,

Thank you for submitting your manuscript to PLOS ONE. After careful consideration, we feel that it has merit but does not fully meet PLOS ONE’s publication criteria as it currently stands. Therefore, we invite you to submit a revised version of the manuscript that addresses the points raised during the review process.

We look forward to receiving your revised manuscript.

Kind regards,

Mohammad Hossein Ebrahimi

Academic Editor

PLOS ONE

Journal Requirements:

Reviewers' comments:

Reviewer's Responses to Questions

**Comments to the Author**

1. Review Comments to the Author

Reviewer #1: Dear, Authors thank you for your response and I hope you will take actions to correct some parts in your manuscript as much as possible. And some comments are attached here

Abstract part

Line-26: This study aims to…. Please! Modify to “This study aimed to….” Or “The aim of this study was….”

Line 31-35: please! Modify to: The collected data were entered into a computer using the software Epidata 3.1 version and exported to SPSS version 23 for statistical analysis. Descriptive and inferential analysis was conducted. Logistic regression analysis was done, and a

statistical association was declared at p-value < 5% and 95% confidence interval(CI).Then the results were presented using a frequency table, figures, and description.

Line 38-39: please! Modify to: Mothers practice of infant sunlight exposure was affected

39 by age of mothers 30–34 years, [AOR= 3.095, 95%CI (1.126, 8.506)], and age ≥35 years, [AOR= 4.491, 95%CI (1.196, 16.863)], and living in urban, [AOR= 1.938, 95%CI (1.053, 3.566)].

Materials and methods part page 6

Line 113-115: please! Modify to: Currently, it is providing emergency services, outpatients’ services, inpatients services, major and minor surgeries, maternal care, pediatrics services, antiretroviral therapy (ART) and Tuberculosis (TB) treatment clinics, laboratory, radiography services, and EPI services for the community.

Sample size Determination part page 6

Line 124-128. The sample was calculated using a 95% confidence interval level, 5% margin of error, and 54.5%: based on this the calculated sample size is

(1.96)2x0.545x (1-0.545)/0.052=381

Please! Correct your sample size calculation. The sample size could be more than 380 as shown above based on your explanation. So how it could be reduced to 220? Please! Write clear explanation about it.

Factors associated with sunlight exposure of infant’s page 13

Line 213-214: Mothers in the age group of 30–34 years and ≥35 were 3.20 [COR= 3.20, 95%CI (1.18, 8.69)],

Also please! Correct sentence of line 220-221 in similar manner

Reviewer #2: The authors may improve the manuscript using the following suggestions:

Abstracts:

1 The authors just need to keep two decimal points for the odds ratio

2 It is too early to suggest “Health professionals should provide health education for mothers at young ages and mothers from rural areas on the benefits of infants' sunlight exposure.” based on one cross-sectional study.

Introduction:

1. Thank you for working on the introduction. If the authors would like to sum up the introduction section concisely, you can use the following detailed information to improve the re-submitted draft.

The authors can

1)start with the outcome (prevalence of rickets in the study area) (The authors can use rows 90-92), and add information about the significance of rickets (mortality, disability, medical cost, et al)

2)state the association between rickets and vitamin D deficiency (The authors can use rows 58-61, 66-67, 78-81, 86)

3)provide the prevalence of vitamin D deficiency in study area (The authors can use rows 89-90)

4)state the sources of vitamin D in infant and the role of sunlight in vitamin D level (The authors can use rows 49-50, 62-65)

5)state sunlight would be the best solution due to the sufficient sunlight in study area, the lack of vitamin D supplements, and the common use of breastfeeding in study area (The authors can use rows 53-57, 68-76, 82, 88, 93, 103)

6)provide the prevalence of sunlight exposure in study area and factors affect the sunlight exposure in study area (The authors can use rows 51, 87, 94-97, 99-102)

7)add articles that indicate the lack of published study on practice of infant sunlight exposure and associated factors among study population

8)the aim of the study to address the knowledge gap (The authors can use rows 104-105).

Please also remove repeated information if the authors choose to rearrange the introduction section using this format.

2. If the authors added rows 94-98 as the knowledge gap to address previous comment #4 (there were no highlights in the introduction), then the authors need to provide why “The current study included the place of residence in addition to previously studied variables.” is the innovation/importance of the current study. Are there any articles indicating that place of residence is an important factor for sunlight exposure? The authors may need to add one sentence here.

3. Other suggestions about the introduction that might improve the introduction:

1) Reference 10 may not support the statement

2) The authors can use “children unexposed to sunlight “ to replace “unexposed children to sunlight”

3) Lack of reference in row 92

Methods:

1 The “10% non-response rate” is not consistent with the “which gives a 100% response rate”

2 Please add the year to the “from 01 June to 30 July.”.

3 The power calculation needs to be improved. 1) The authors can use a fixed number of participants to calculate power, or the authors can calculate the number of participants required to reach the 80% statistical power at the significance level of 0.05. The authors can also calculate the minimal detectable odds ratios using the fixed number of participants and the 80% statistical power. 2) The information provided in the sample size determination section is not sufficient to calculate sample size or power using the Epiinfor 7 software.

Results:

1 “More than two-thirds (69.1%) of the mothers said that morning is a good time for infants’ sunlight exposure (Table 3).” is not consistent with the “99.1% “ reported in table 3.

2 The percentages for “Fear to expose baby on sunlight” in Table 4 are not consistent with the frequency.

Discussion:

1 For the first two paragraphs, please just mention the important findings. Please try not to repeat what you had in the result section.

2 Please revise the “The study could not support the cause and effect relationship between independent and dependent variables.” We usually would not use the terms “cause” and “effect” in the cross-sectional study.

Reviewer #3: Data analysis procedure: Please describe the variables included in the logistic regression analyses and define AOR.

Results: the factors associated with sunlight exposure of infants, which were found in logistic regression, are unclear. The reader needs to know the coefficients, and you must define what COR is in the first mention.

Include in all tables notes defining all the acronyms used.

---

## [Author Response · Author response to Decision Letter 1]

4 Oct 2022

Responses to Editor(s) and Reviewers’ comments 

Manuscript Id: PONE-D-21-36839R1

Title: Factors Associated with Infants’ Sunlight Exposure among Mothers Attending EPI of Wolkite University Specialized Hospital

Dear editor(s) and Reviewers I would like to say thank for your comments and recommendations. I have included the concerns raised by reviewers in the documents, and if there is untouched points I am ready to amend. 

Thank you for reviewing the paper. 

Section Comments Responses

Journal Requirements:

 Please review your reference list to ensure that it is complete and correct. Upon reviewing the references, three of them were found incorrectely written and modified.

 Reviewer #1 

Abstract part Line-26: This study aims to…. Please! Modify to “This study aimed to….” Or “The aim of this study was….” Modified 

 Line 31-35: please! Modify to: The collected data were entered into a computer using the software Epidata 3.1 version and exported to SPSS version 23 for statistical analysis. Descriptive and inferential analysis was conducted. Logistic regression analysis was done, and a

statistical association was declared at p-value < 5% and 95% confidence interval(CI).Then the results were presented using a frequency table, figures, and description. Modified 

 Line 38-39: please! Modify to: Mothers practice of infant sunlight exposure was affected by age of mothers 30–34 years, [AOR= 3.095, 95%CI (1.126, 8.506)], and age ≥35 years, [AOR= 4.491, 95%CI (1.196, 16.863)], and living in urban, [AOR= 1.938, 95%CI (1.053, 3.566)]. Modified 

Materials and methods part page 6 Line 113-115: please! Modify to: Currently, it is providing emergency services, outpatients’ services, inpatients services, major and minor surgeries, maternal care, pediatrics services, antiretroviral therapy (ART) and Tuberculosis (TB) treatment clinics, laboratory, radiography services, and EPI services for the community. Modified 

Sample size Determination part page 6

 Line 124-128. The sample was calculated using a 95% confidence interval level, 5% margin of error, and 54.5%: based on this the calculated sample size is

(1.96)2x0.545x (1-0.545)/0.052=381

Please! Correct your sample size calculation. The sample size could be more than 380 as shown above based on your explanation. So how it could be reduced to 220? Please! Write clear explanation about it. We changed the methods sample size calculations based on your concern first we used epi info, and due to error in metnod of calculation we used correction formula.

Factors associated with sunlight exposure of infant’s page 13 Line 213-214: Mothers in the age group of 30–34 years and ≥35 were 3.20 [COR= 3.20, 95%CI (1.18, 8.69)],

Also please! Correct sentence of line 220-221 in similar manner Modified 

 Reviewer #2: 

Abstracts:

 1 The authors just need to keep two decimal points for the odds ratio Corrected to two decimal place 

 2 It is too early to suggest “Health professionals should provide health education for mothers at young ages and mothers from rural areas on the benefits of infants' sunlight exposure.” based on one cross-sectional study. Modified 

Introduction: 1. Thank you for working on the introduction. If the authors would like to sum up the introduction section concisely, you can use the following detailed information to improve the re-submitted draft.

The authors can

1)start with the outcome (prevalence of rickets in the study area) (The authors can use rows 90-92), and add information about the significance of rickets (mortality, disability, medical cost, et al)

2)state the association between rickets and vitamin D deficiency (The authors can use rows 58-61, 66-67, 78-81, 86)

3)provide the prevalence of vitamin D deficiency in study area (The authors can use rows 89-90)

4)state the sources of vitamin D in infant and the role of sunlight in vitamin D level (The authors can use rows 49-50, 62-65)

5)state sunlight would be the best solution due to the sufficient sunlight in study area, the lack of vitamin D supplements, and the common use of breastfeeding in study area (The authors can use rows 53-57, 68-76, 82, 88, 93, 103)

6)provide the prevalence of sunlight exposure in study area and factors affect the sunlight exposure in study area (The authors can use rows 51, 87, 94-97, 99-102)

7)add articles that indicate the lack of published study on practice of infant sunlight exposure and associated factors among study population

8)the aim of the study to address the knowledge gap (The authors can use rows 104-105). We leave the introduction part as it is. The flow of ideas were started from benefits of sunlight exposure, vitamin D and sunlight exposure, deficiency of vit D, its epidemiology, relation to exposure, … 

This is the reason for no rearranging the introduction. 

But, we appreciate your suggestions. 

 Please also remove repeated information if the authors choose to rearrange the introduction section using this format. We decided no to rearrange the part. 

 2. If the authors added rows 94-98 as the knowledge gap to address previous comment #4 (there were no highlights in the introduction), then the authors need to provide why “The current study included the place of residence in addition to previously studied variables.” is the innovation/importance of the current study. Are there any articles indicating that place of residence is an important factor for sunlight exposure? The authors may need to add one sentence here. Of course no, but we were interested to study it, from our clinical experience. 

3. Other suggestions about the introduction that might improve the introduction: 1) Reference 10 may not support the statement Removed 

 2) The authors can use “children unexposed to sunlight “ to replace “unexposed children to sunlight” Replaced 

 3) Lack of reference in row 92 Revised and added.

Method: :1 The “10% non-response rate” is not consistent with the “which gives a 100% response rate” you are right, but all distributed paper was returned from data collectors. When we check the questionnaire all were complete and decicded to include all. 

 2 Please add the year to the “from 01 June to 30 July.”. Added 

 3 The power calculation needs to be improved. 1) The authors can use a fixed number of participants to calculate power, or the authors can calculate the number of participants required to reach the 80% statistical power at the significance level of 0.05. The authors can also calculate the minimal detectable odds ratios using the fixed number of participants and the 80% statistical power. 2) The information provided in the sample size determination section is not sufficient to calculate sample size or power using the Epiinfor 7 software. From your comments we understand that we wrongly used epi info based sample size calculation. To make clarity, we decided to use correction formula since the population was small. 

Results: 1 “More than two-thirds (69.1%) of the mothers said that morning is a good time for infants’ sunlight exposure (Table 3).” is not consistent with the “99.1% “ reported in table 3. Corrected 

 2 The percentages for “Fear to expose baby on sunlight” in Table 4 are not consistent with the frequency. Multiple resposnes are possible. 

Discussion:

 1 For the first two paragraphs, please just mention the important findings. Please try not to repeat what you had in the result section. Reviewed and modified 

 2 Please revise the “The study could not support the cause and effect relationship between independent and dependent variables.” We usually would not use the terms “cause” and “effect” in the cross-sectional study. We removed the the sentence 

 Reviewer #3 

Data analysis procedure Please describe the variables included in the logistic regression analyses and define AOR. Described, and defined 

Results the factors associated with sunlight exposure of infants, which were found in logistic regression, are unclear. The reader needs to know the coefficients, and you must define what COR is in the first mention. Included 

 Include in all tables notes defining all the acronyms used. Included

---

## [Editor Report · Decision Letter 2]

10 Oct 2022

PONE-D-21-36839R2Factors associated with infants’ sunlight exposure among mothers attending the EPI unit of Wolkite University Specialized HospitalPLOS ONE

Dear Dr. Mengistu,

Thank you for submitting your manuscript to PLOS ONE. After careful consideration, we feel that it has merit but does not fully meet PLOS ONE’s publication criteria as it currently stands. Therefore, we invite you to submit a revised version of the manuscript that addresses the points raised during the review process.

We look forward to receiving your revised manuscript.

Kind regards,

Mohammad Hossein Ebrahimi

Academic Editor

PLOS ONE
---

## [Author Response · Author response to Decision Letter 2]

23 Oct 2022

Journal Requirements:

 Please review your reference list to ensure that it is complete and correct. We reviewed and checked the references, all are correct and available online.

---

## [Editor Report · Decision Letter 3]

26 Oct 2022

Factors associated with infants’ sunlight exposure among mothers attending the EPI unit of Wolkite University Specialized Hospital

PONE-D-21-36839R3

Dear Dr. Mengistu,

We’re pleased to inform you that your manuscript has been judged scientifically suitable for publication and will be formally accepted for publication once it meets all outstanding technical requirements.

Kind regards,

Mohammad Hossein Ebrahimi

Academic Editor

PLOS ONE
---

## [Editor Report · Acceptance letter]

9 Nov 2022

PONE-D-21-36839R3 

Factors associated with infants’ sunlight exposure among mothers attending the EPI unit of Wolkite University Specialized Hospital 

Dear Dr. Mengistu:

I'm pleased to inform you that your manuscript has been deemed suitable for publication in PLOS ONE. Congratulations! Your manuscript is now with our production department. 

Kind regards, 

on behalf of

Dr. Mohammad Hossein Ebrahimi 

Academic Editor

PLOS ONE